# Datasets for Navigating Sensitive Topics in Preference Data and Recommendations

**Amelia Kovacs**
Cornell University
Ithaca, NY
ajk296@cornell.edu

**Jerry Chee**
Cornell University
Ithaca, NY
jc3464@cornell.edu

**Kimia Kazemian**
Cornell University
Ithaca, NY
kk983@cornell.edu

**Sarah Dean**
Cornell University
Ithaca, NY
sdean@cornell.edu

## Abstract

Personalized AI systems, from recommendation systems to chatbots, are a prevalent method for distributing content to users based on their learned preferences. However, there is growing concern about the adverse effects of these systems, including their potential tendency to expose users to sensitive or harmful material, negatively impacting overall well-being. To address this concern quantitatively, it is necessary to create datasets with relevant sensitivity labels for content, enabling researchers to evaluate personalized systems beyond mere engagement metrics. To this end, we introduce two novel datasets that include a taxonomy of sensitivity labels alongside user-content ratings: one that integrates MovieLens rating data with content warnings from the Does the Dog Die? community ratings website, and another that combines fan-fiction interaction data and user-generated warnings from Archive of Our Own. We conduct comprehensive summary statistical analyses on these datasets and train three distinct recommendation algorithms on each. Our experimental analysis examines how these algorithms either amplify or mitigate the presence of content warnings. This work aims to provide critical insights into whether standard recommendation systems disproportionately highlight sensitive content and offers a robust foundation for future research and the development of more nuanced AI systems that account for content sensitivities.

## 1 Introduction

Over the past decade there has been growing concern in academic, public, and regulatory spaces about negative effects of personalized recommendation. For example, fertility related advertisements have been repeatedly shown on YouTube to infertile women whom have tried to opt out of these advertisements (Vox, 2020), and eating disorder content has been found to be algorithmically recommended on TikTok (Mashable, 2020). Advances in other personalized AI systems, like chatbots, are posed to further this problem. These AI systems are classically designed to maximize engagement, and there has been significant work investigating how this objective may lead to the aforementioned adverse outcomes. Some concerns are societal-level (e.g. polarization, filter bubbles), some are mediated by individuals (e.g. radicalization, conspiracy), and some are focused on the individual level (e.g. bias, sensitive content). It is challenging to study the relationship between societal-level phenomena and personalization because the phenomena result from complex social forces. On the other hand, harms at the individual level should be more easily addressed.

38th Conference on Neural Information Processing Systems (NeurIPS 2024).

For a research community focused on personalization, it is important to address the problem of individual level harms. While academic work has investigated bias and fairness, little work has focused on the question of sensitive or harmful content. The nature of the existing academic work on harm has typically been theoretical, proving properties of simplified mathematical models, or audit based, by creating fake profiles on social media sites and tracking recommendations. We present an extensive discussion of related work in Appendix A.

In this paper, we address the need to better understand the interplay between preference data, personalization, and sensitive or harmful content. Benchmark datasets are crucial to enable such analysis and promote the development of better personalization strategies. Our work supports this motivation by proposing novel datasets that augment standard preference data to account for certain notions of harm. *Sensitivity labels*—including trigger warnings or content warnings—have naturally emerged on many online media platforms. They provide a categorization system which allows different users to avoid subsets of content they find objectionable. These labels provide an explicit way to measure a form of user harm. We emphasize that sensitive content shouldn't be outright banned; instead, users should have the agency to avoid such content, if they choose. For example, some users may choose to avoid animal deaths in their content recommendations, so measuring the prevalence of animal deaths in recommendations provides insights into potential user harm. Our proposed datasets with sensitivity data will allow the research community evaluate if personalization with preference data drives increased harm measured via content warnings.

Our contributions are: *First,* we propose two novel datasets which augment standard user-content preferences with content warning labels. The first dataset augments the MovieLens rating datasets with warnings from `doesthedogdie.com`. The second dataset consists of fan-fiction interactions from `archiveofourown.org` and warnings from the Webis Trigger Warning Corpus (Wiegmann et al., 2023). *Second,* we conduct descriptive analyses on our novel datasets to understand the distribution of these sensitive labels with respect to user-content interactions. *Third,* we demonstrate the value of this data by presenting a preliminary analysis investigating the extent to which standard recommendation algorithms amplify the prevalence of sensitive labels. Our initial findings indicate that personalized recommendation algorithms do not amplify sensitive content, compared with non-personalized popularity-based and random recommendations. *Finally,* we make our datasets [1] and code [2] publicly available for other researchers to build on.

Section 2 describes our proposed datasets and summary analysis. In Section 3 we conduct an experimental analysis on how standard recommendation algorithms affect amplification of sensitive content. Section 4 concludes with a discussion and motivation for future work.

## 2   Datasets

We present two datasets to enable investigation of the relationship between sensitive content and recommendation systems. The datasets each contain two tables: 1) A *sensitivity table* enumerating item identifiers and associated content warnings, and 2) An *interaction table* listing the rating or presence of a "like" between each user and item in the dataset. Instructions and code to download and utilize these datasets can be found here. We have either obeyed scraping limits, or received permission to collect data from our sources.

### 2.1   MovieLens and Does the Dog Die?

MovieLens is a movie recommendation service which provides data on user ratings of movies (Harper and Konstan, 2015). They provide several publicly available datasets which are widely used to study recommendation algorithms. User ratings range from 0.5 (dislike) to 5 (like), representing an explicit interaction. To study the relationship between user preferences and sensitive content, we augment the Movielens 25M dataset (Harper and Konstan, 2015) with additional information about each of the movies. We leverage `doesthedogdie.com` (DDD), a platform of community-generated trigger warnings for movies, TV shows, and other media. Users are able to vote "Yes" or "No" on whether a

---

[1] Relevant data files can be found on Hugging Face: `https://huggingface.co/datasets/sdeangroup/NavigatingSensitivity`

[2] We provide code and instructions to download, clean, process, and analyze the data on our GitHub: `https://github.com/sdean-group/Navigating-Sensitivity`

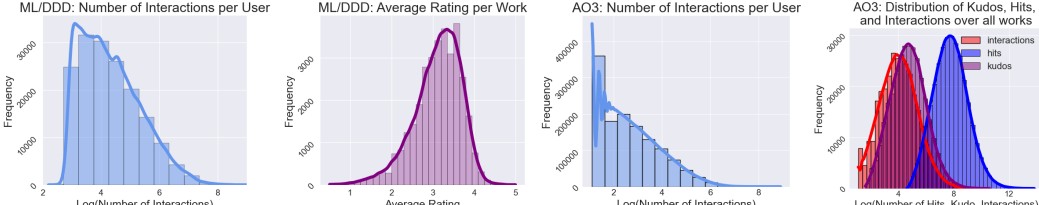

Figure 1: (Left) Distribution of the average rating per work in the MovieLens/Does the Dog Die? dataset. (Middle Left) Distribution of the number of interactions per user in the MovieLens/Does the Dog Die? dataset. (Middle Right) Distribution of the number of interactions per user in the AO3 dataset. (Right) Distribution of the number of hits (reads), kudos (likes), and interactions per work in the AO3 dataset.

given warning applies to a work, providing a user-driven method of determining sensitivity labels. Using the DDD API, we match the IMDb and TMBD identifiers of movies in the MovieLens 25M dataset to entries on DDD to create a sensitivity table of movies and their associated content warnings. We create our user interaction table by filtering the rating dataset provided by MovieLens to only contain works found on DDD (52% of the movies) and users who interacted with at least three works in the sensitivity table (100% of users). Moving forward we will refer to this dataset as ML-DDD.

*Summary Statistics.* Our dataset contains 32,604 movies, 162,541 users, and 22,867,672 interactions. On average, the movies receive 701.37 ratings and a mean rating of 3.11. There are 137 distinct content warnings in DDD. A full list is provided in the appendix. We consider a work as having a content warning (Clear Yes) if at least 75% of the votes are "Yes". Similarly, a work does not have the warning (Clear No) if at least 75% of the votes are "No". The sensitivity table has a column for each of these categories for each warning and are hot-encoded to indicate if a warning label is present. In total, there are 155,852 "Clear Yes" warning labels, 761,303 "Clear No", 36,080 "Unclear", and 3,513,512 "No Votes". Given the large number of "Clear No" consensuses, we compare works with and without a warning via the "Clear Yes" and "Clear No" labels throughout our analysis.

Additionally, we define two metrics for density: *Interaction Density* is the number of interactions divided by the product of the number of users and items. *Warning Density* is the number of instances that a warning is applied to a work divided by the number of warnings times the number of works. The dataset has an interaction density of 0.43% and a warning density of 3.49%.

## 2.2  Archive Of Our Own

Archive Of Our Own (AO3) is a massive repository of fan-fictions written and read by millions of users. The works and their metadata are publicly available, such as the number of reads ("hits"), the number of likes ("kudos"), and the names of public user accounts who have given kudos.

What makes the archive particularly unique is the thorough tagging system maintained by users of the site. Works are tagged with user-generated labels describing the contents of the work and often serve as a warning for potentially triggering or harmful content. Wiegmann et al. (2023) systematically categorize 41 million user-generated tags into 36 different trigger warning categories. These categories were selected based on a literature review of institutionally recommended warnings. They propose the Webis Trigger Warning Corpus, a dataset of 1 million fanfiction works, metadata, and corresponding trigger warnings.

To use this data in recommendation systems it is essential to obtain information on user interactions with works, which is lacking from the Webis Trigger Warning Corpus. To collect this data, we modify the data collection process to collect the publicly available usernames of those who gave kudos to each work, as well as updated kudos and hit metrics. Giving kudos to a work represents an implicit interaction: we only know what works a user has liked, and cannot assume a user does not like unseen works. It is worth noting that the number of identifiable users who give public kudos does not always equal the total number of kudos, as individuals are able to interact as guests.

We collected[3] interactions for a subset of works from the Webis Trigger Warning Corpus (nearly 30%) between March, 2024, and May, 2024. Our dataset excludes works works with less than three

---

[3]When performing this collection we adhered by the scraping rate limits enforced by AO3.

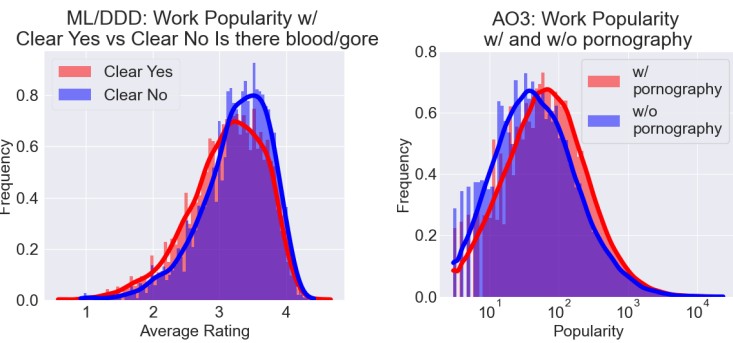

Figure 2: (Left) Distribution shift of average ratings per work with and without the "Is there blood/gore" label in ML-DDD. (Right) Distribution shift of the number of public kudos (interactions) received by works with versus without pornography in AO3.

publicly available user interactions and kudos by users who have interacted with less than three works in the dataset. Additionally, our data has pseudonymized the usernames with unique numerical identifiers to respect the privacy of AO3 users. Moving forward we will refer to this dataset as AO3.

*Summary Statistics.* Our dataset contains 306,111 works, 1,304,303 users, and 45,936,871 interactions between users and works. On average, works receive 5,468.40 hits, 265.83 kudos, and 150.07 kudos from publicly identifiable users. We will refer to this last metric simply as "interactions".

The Webis Trigger Warning Corpus taxonomizes warning labels into 4 categories: fine open, fine closed, coarse open, and coarse closed. We examine the 36 distinct fine closed warnings. A full list of the labels and their respective prevalence is included in the appendix. Given the thorough categorization performed by Wiegmann et al. (2023), we consider works marked by a warning similarly to the "Clear Yes" category of ML-DDD and those without a warning as "Clear No". The AO3 sensitivity table contains a column for each warning with a 1 or 0 to indicate their presence or absence. In total there are 579,610 work-warning pairs where the warning is present in the work. The dataset has an interaction density of 0.012% and warning density of 5.26%, calculated as described in Section 3.1. In comparison to the ML/DDD dataset, the AO3 dataset is larger, sparser in interactions, denser in warnings, and contains implicit rather than explicit user interactions.

## 2.3 Sensitivity Label Analysis

We begin by presenting a broad overview of the relationship between sensitive content and user preferences. This investigation of the relationship between sensitivity and popularity contextualizes our later experimental analysis of sensitive labels and recommendation. We consider popularity in two ways, aimed at gauging the quality of content. Movies in ML-DDD are measured by their average rating, and works in AO3 are measured by the number of kudos received from publicly identifiable users ("interactions" in the dataset).

First, we investigate the existence of a relationship between sensitivity and popularity across various sensitive categories. We seek to answer the following question: does the distribution of popularity for works marked with a content warning match the distribution of popularity for works without the warning? The presence and absence of warnings in ML/DDD is indicated by the "Clear Yes" and "Clear No" markings respectively for each label. In AO3, we directly compare the presence or absence of a warning label across works.

We compare the popularity distributions visually by plotting histograms of the average rating (ML/DDD) or number of interactions (AO3) in works with or without a given warning label. Figure 2 displays two examples of these histograms for the most prevalent warnings in each dataset: "Is there blood/gore" in ML/DDD and "pornography" in AO3. Visually, we can see a clear shift in both the distributions.

To summarize the distribution shifts quantitatively, we perform a permutation test to examine the following null hypothesis: the presence of a content warning has no correlation with work popularity. Figure 3 shows the results of this test for the top 10 most prevalent content warnings in the ML/DDD

and AO3 datasets. The direction of the bars indicates how the presence of the label correlates with popularity: positive indicates works with the given label receive higher ratings or more interactions than expected under the null hypothesis, and negative indicates less. 70% of the ML/DDD labels and only 50% of the AO3 have a negative correlation, meaning the collection of works with those warnings are systematically less popular.

We also compute $p$ values to determine the statistical significance of these results. 70% of both the ML/DDD and AO3 labels have $p$ values below 0.005. Rejecting the null hypothesis, we conclude that there is a correlation between those labels and work popularity. We expand this analysis to all sensitivity labels in the two datasets in our provided code repository.

Next, we aim to understand how individual users interact with content given the presence of warnings. We selected 1,000 random users from each dataset for the following analysis. Figure 4 plots the average rating (ML/DDD), as well as the number of interactions (AO3) for each user. The x-axis plots a user's average rating or number of interactions for works with "Is there blood/gore" or "pornography", and the y-axis plots a user's average rating or number of interactions for works without said sensitivity label. The diagonal red line has slope of one, indicating a user has equal average ratings for works with and without said sensitively label. 57.30% of the ML/DDD users fall below the red line, indicating they rate movies with the "Is there blood/gore" label higher. The result of the label's permutation test (Figure 3) indicates works with the "Is there blood/gore" content warning are rated lower on average, highlighting how individual user preferences do not always align with global popularity trends.

Figure 4 shows how AO3 users tend to interact more with content labeled "pornography", as indicated by the 72.60% of users who fall below the red line. Unlike the ML/DDD example, this finding is in line with the positive observed difference from the permutation test (Figure 3).

## 3 Initial Amplification Analysis

The primary purpose of our novel datasets is to enable the training and evaluation of recommender systems with the notion of sensitive or harmful content in mind. To lay the foundation for this work, we perform a preliminary analysis of three different recommendation algorithms on each dataset. The recommenders are trained only on interaction data and then evaluated on their performance predicting ratings or recommendations and their amplification of the sensitivity labels.

We evaluate accuracy metrics of two baselines and one personalized recommendation algorithm on a 90-10 train-test split of the data. We then train the models on 100% of the interactions and evaluate the amplification of warnings in the top 100 recommendations for 1,000 randomly chosen users. Further details of training and the resulting accuracy metrics are presented in Appendix C

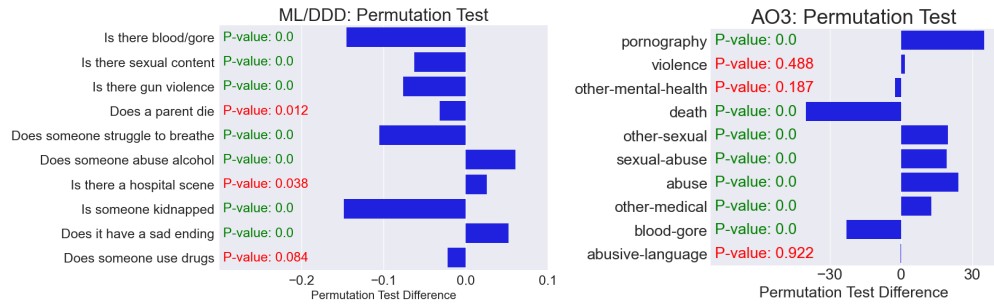

Figure 3: (Right) Permutation test results for the top 10 most prevalent AO3 content warnings. The permutation test examines if the presence of a content warning has a correlation with the number of interactions received by works. The direction of the bars indicates whether the correlation is positive or negative, and the p-values determine statistical significance. 70% of the labels have p-value less than 0.005. (Left) Permutation test results for the top 10 most prevalent ML/DDD content warnings. The permutation test examines if the presence of a content warning has a correlation with work average ratings. The direction of the bars indicates whether the correlation is positive or negative, and the p-values determine statistical significance. 70% of the labels have p-value less than 0.005.

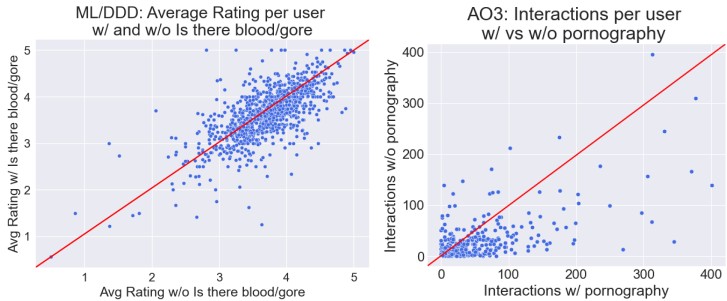

Figure 4: (Left) A per-user visualization of average ratings for works with versus without the blood/gore warning in ML/DDD. Each point represents one of 1,000 random users. 57.30% of the users give higher average ratings to movies with the warning. (Right) A per-user visualization of the number of public kudos (interactions) received for works in AO3 with versus without the pornography warning. Each point represents one of 1,000 random users. 72.60% of the users have given more kudos to works with the warning.

## 3.1 Algorithms

We consider two non-personalized recommendation algorithms, Random and TopPop, as baselines for each dataset (Kille and Lommatzsch, 2019). The Random algorithm generates recommendations at random, drawing works a user has not interacted with from a uniform distribution. The TopPop algorithm sorts works by popularity and recommends the top-k that a user has not seen. Popularity is measured by the average rating for a movie in ML-DDD and the number of interactions with a work in AO3.

For a more personalized approach, we also trained a Collaborative Filtering algorithm on each dataset. Collaborating filtering is a technique used in recommendation systems to predict the preferences of a user by ammassing data on other user-item interactions. It assumes that similar users will prefer similar items and uses this principle in generating ratings or recommendations.

Singular Value Decomposition (SVD) is a well-established matrix factorization technique (Mnih and Salakhutdinov, 2007). It performs collaborative filtering by constructing matrices of user and item latent factors to generate predictions. We apply SVD to the ML/DDD dataset using the Python library Surprise (Hug, 2020).

Unlike the explicit 0.5-5 scale ratings of ML-DDD, the AO3 dataset contains implicit user interactions from publicly available accounts awarding Kudos to a work. This means there is no clear notion of dislike in the dataset as it is not differentiated from works that a user simply never came across. For this reason, we use the Implicit (Frederickson, 2023) Python library's implementation of an Alternating Least Squares (ALS) (Hu et al., 2008) algorithm catered for implicit data to generate recommendations for the AO3 dataset. This algorithm also uses matrix factorization for collaborative filtering, but deals with missing values by iteratively learning confidence levels to determine if the missing values indicate a negative or positive preference.

## 3.2 Results

When considering the relationship between recommendation systems and content warnings, we introduce a novel metric called *Warning Amplification* inspired by metrics for algorithmic amplification (Huszár et al., 2021; Bouchaud, 2024). $Amplification@k$ is define as: for a given user, the fraction of items in their k top recommendations which have a given warning divided by the fraction of items in their history with the warning. We calculate this metric for each user in the randomly selected subset. We normalize this metric by subtracting 1, so 0% amplification indicates the user receives the same amount of warnings in their recommendations as exists in their history. To avoid dividing by zero in the case when the user does not have any items with the warning in their history, we hallucinate a single item with the given warning in the user's history.

We present the distributions of user average amplification scores for k = 100 recommendations generated by each of the algorithms in Figure 5 (ML-DDD) and Figure 5 (AO3). A user's average

amplification is calculated by computing the mean of their Amplification@k scores for each warning. The black dashed line indicates where amplification is 0%, i.e. there is no difference in the amount of warnings in a user's relevant history and their recommendations. In ML-DDD, we define relevant works as those in a user's top quartile of ratings. In AO3, all works in a user's history are relevant since they correspond to a positive interaction (kudos).

In Figure 5 we see that TopPop produces the most amplification and SVD the least. TopPop recommends popular works to users, and as examined in the permutation tests, a high proportion of warnings correlate positively with work popularity. Given the non-personalized nature of this algorithm, users are given popular yet warning-dense recommendations. The Random recommender chooses works at random, and given the low warning density of the dataset as noted in Section 3.1, it is not surprising that the algorithm produces less recommendations with warnings than those in a user's relevant history. The SVD recommender has the lowest average amplification scores and recommends less items with a warning than in a user's relevant history.

In Figure 5 we see the user average amplifications for AO3. The Random algorithm produces the most amplification and ALS the least. The Random algorithm likely amplifies warnings more than in ML-DDD given the higher warning density of the AO3 dataset. Similarly to the ML-DDD analysis, the personalized recommender amplifies warnings the least, consistently deamplifying sensitive content.

We also consider the average amplification of each warning, calculated by computing the mean of all user's Amplification@k scores for the single warning. In Figure 6 we see a comparison of the amplification of "Is there blood/gore" by the ML-DDD algorithms and "pornography" by the AO3 algorithms. Following suit with the user average amplification analysis for ML-DDD, TopPop produces the most amplification of the blood/gore warning and SVD the least. The user average amplification trends for AO3 are also closely replicated on the pornography warning, but with TopPop amplifying slightly more than Random. From the permutation tests, we know blood/gore correlates negatively with popularity and pornography correlates positively. This is reflected in the figures as amplification for blood/gore for all algorithms is much lower than that of pornography.

Overall, our preliminary analysis of the relationship between sensitive categories and recommendation systems reveals how the personalized algorithms do not amplify warnings in comparison to baselines. This finding presents an interesting avenue for future research to explore, aiming to understand the nuances in the algorithms that lead to deamplification.

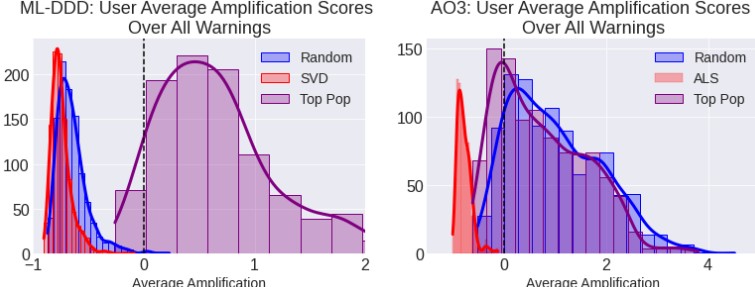

Figure 5: (Left) Distribution of amplification scores averaged over all warnings in ML-DDD for each of the 1,000 randomly selected users. Amplification is the ratio of works in a user's recommendations and their history which have a given warning. The black dashed line represents no amplification. TopPop produces the most amplification while SVD produces the least. (Right) Distribution of amplification scores averaged over all warnings in AO3 for each of the 1,000 randomly selected users. Amplification is the ratio of works in a user's recommendations and their history which have a given warning. The black dashed line represents no amplification. Random produces the most amplification while ALS produces the least.

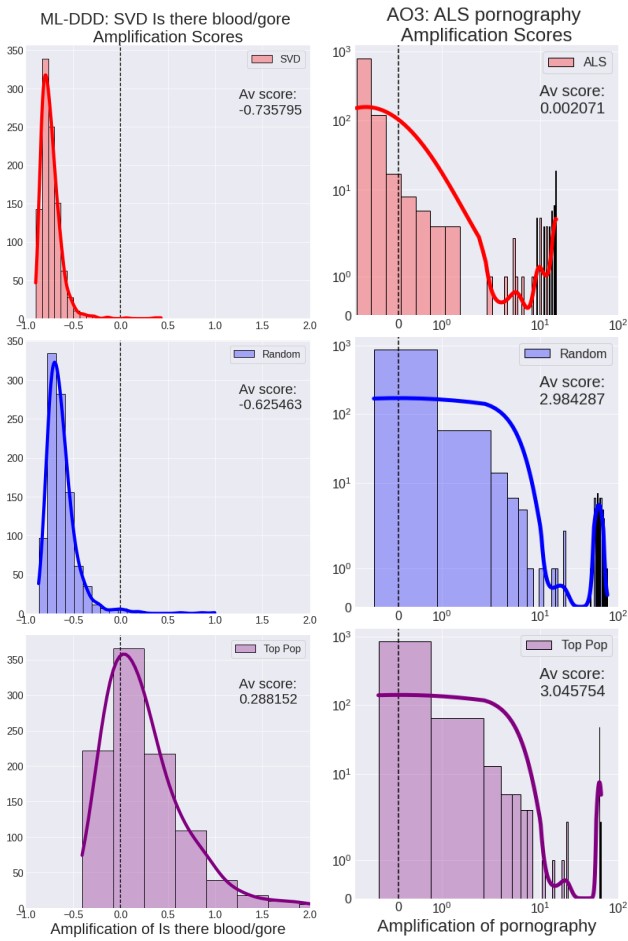

Figure 6: Amplification of the ML-DDD "Is there blood/gore" warning by the Random, TopPop, and SVD algorithms for k = 100 recommendations. The black dashed line represents 0% amplification, indicating the percent of works with blood/gore is the same in a user's history and their recommendations.

## 4  Discussion and Future Work

In this paper, we present two novel recommendation datasets, ML-DDD and AO3, which include both user-content interaction data as well as information about sensitive content categories. We present descriptive analyses of both datasets to understand the relationship between popularity and sensitivity. These analyses contextualize our preliminary analysis of amplification: we train three recommendation algorithms on the interaction data, and then evaluate the extent to which content warnings are amplified. Interestingly, we find that personalized recommendation algorithms do not amplify sensitive content, especially compared to non-personalized popularity based algorithms, but also even compared with random recommendations.

There remain many opportunities for future work. One important direction is to extend our preliminary analysis to understand how (and whether) recommendation algorithms amplify sensitive content. Another interesting set of questions arises from sensitive label disagreement, for example by investigating disagreement in user votes in DDD. Lastly, there are many open questions around how to design recommendation algorithms that take sensitive content seriously; for example by modelling negative preferences or providing richer user controls. It is important to address the individual-level harms that occur when unwanted sensitive content is recommended. We hope that these datasets will spur the research community to tackle this problem.

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

# A   Related Work

## A.1   Datasets of Sensitive or Harmful Content

There are several types of sensitive or harmful content, and several academic datasets for studying their prevalence online. Content which spreads misinformation or conspiracy theories has received much attention. Faddoul et al. (2020) study the presence of conspiracy videos on YouTube over time, and Liaw et al. (2023) provide the YouNICon dataset of conspiracy YouTube videos. Nielsen and McConville (2022) provide Mumin, a dataset of fact checked misinformation on Twitter. This type of harmful content has also recieved attention from the recommender systems community: Fernández et al. (2021) and Tommasel and Menczer (2022) link misinformation labels with recommendation data to study amplification.

Another type of harmful content is hate speech or toxic content. Almerekhi et al. (2020) present a dataset of Reddit comments and toxicity labels. Mollas et al. (2020) present a dataset of hate speech in Reddit and YouTube comments. Wulczyn et al. (2017) provide a dataset of personal attacks in Wikipedia comments. For all of these works, the labels were generated by crowd workers.

The aforementioned types of harmful content are generally viewed as universally negative, and thus datasets are created and used with the implicit goal of limiting or preventing its proliferation. These goals can be fraught, since the precise definition of concepts like "hate speech" and "misinformation" are often contested and may suffer from biases against marginalized groups (Vidgen and Derczynski, 2020).

In contrast, our focus is on user generated content warnings, which are created by the same communities who use them as a tool to curate media consumption. As a result, this form of sensitive content is less relevant to censorship and more relevant to ensuring user agency. Wiegmann et al. (2023) provide the Webis Trigger Warning Corpus, an extensive dataset and taxonomy of trigger warnings which we leverage. Despite the relevance of sensitive content labels to real world recommendation systems like Instagram (Instagram, 2021) and TikTok (Goodman, 2020), we are not aware of recommendation focused datasets that provide sensitive content labels. We hope this work will fill that gap.

## A.2   Harm in Recommendation Systems

There is extensive work in the literature characterizing various types of harmful impacts from recommender systems. Shelby et al. (2023) propose a taxonomy of sociotechnical harms based on an extensive literature review. Five major types of sociotechnical harms are categorized: representational, allocative, quality of service, interpersonal, and social systems harms.

Some harms studied act at the societal level. Ribeiro et al. (2020) conduct an audit study of YouTube to study radicalization pathways. Ledwich et al. (2022) study filter bubbles on YouTube by analyzing the recommendations of stylized bots with content preferences and watch history. Levin et al. (2021) present several works on the dynamics of political polarization; mathematical models of polarization are formulated and theoretically analyzed. Restrepo et al. (2021) observe how Facebook parenting communities are pushed closer to extreme communities, and then use a dynamical systems model to derive strategies for controlling this type of amplification. Whittaker et al. (2021) conduct an empirical analysis of YouTube, Reddit, and Gab's recommendation systems when interacting with far-right content, showing that YouTube amplifies extreme content, while Reddit and Gab do not. Gormann and Armstrong (2022) conducts theoretical and simulation studies on the consequences of failed alignment with human values.

Other harms studied act at the individual level. Lin et al. (2016) survey adults age 19 to 32 about social media use and depression, finding that increased social media use correlates strongly with an increased odds of depression. Smith et al. (2022) use case studies to summarize common causes of algorithmic harm and their negative consequences.

Work on multi-objective recommender systems has in part been developed as a response to these observed harms. Zheng and Wang (2022) and Jannach (2022) provide surveys of this field. In addition to the standard engagement maximization, these multi-objective works also optimize for diversity (Vargas and Castells, 2011), fairness (Xiao et al., 2017), multi-stakeholder utility (Sürer et al., 2018), polarization (Suna and Nasraouia, 2021), and harm proxies (Singh et al., 2020). This line of work requires supplemental data to measure the additional objective that is co-optimized with

engagement maximization. The datasets we provide will enable multi-objective work that balances harm as measured by unwanted exposure to sensitive content.

### A.3 Evaluating Fairness, Bias, Amplification

Unfairness and bias are additional negative consequences that may result from recommendation systems. These topics have received much attention from the academic community; we refer to the surveys by Chen et al. (2023) and Ekstrand et al. (2022).

Many measures of bias and (un)fairness capture discrepancies between expressed user preferences and recommendations. Over- or under-recommending certain content categories constitutes *mis-calibrated* recommendations (Steck, 2018). Several works identify bias relating to item popularity (Abdollahpouri et al., 2019; Ekstrand et al., 2018a), item genre (Lin et al., 2019), and creator or user demographics (Ekstrand et al., 2018b,a; Shakespeare et al., 2020) in domains including movies, music, books. Another line of work investigates bias in terms of accessibility using counterfactual metrics. Rather than measuring mis-calibration in what is recommended, it is measured by what *could be* recommended (Akpinar et al., 2022; Dean et al., 2020; Curmei et al., 2021; Guo et al., 2021).

We hope that the datasets we provide will allow for investigations of bias and amplification as they relate to sensitive content; we present a preliminary analysis along these lines in Section 3.

# B  Additional Information

Table 1: The 36 warning labels used in the AO3 dataset and the number of works marked with the warning.

| Warning | Works W/ |
|---|---|
| pornography | 174391 |
| violence | 42084 |
| other-mental-health | 40560 |
| death | 36123 |
| other-sexual | 35784 |
| sexual-abuse | 35005 |
| abuse | 32111 |
| other-medical | 21428 |
| blood-gore | 19299 |
| abusive-language | 18708 |
| suicide | 13113 |
| childbirth | 12132 |
| child-abuse | 11449 |
| mental-illness | 11344 |
| addiction | 11179 |
| incest | 9864 |
| homophobia | 8392 |
| self-harm | 7628 |
| kidnapping | 7561 |
| other-aggression | 5717 |
| collective-violence | 5351 |
| procedures | 4784 |
| dysmorphia | 3525 |
| other-pregnancy | 1984 |
| other-abuse | 1613 |
| sexism | 1606 |
| other-discrimination | 1595 |
| racism | 1159 |
| miscarriage | 930 |
| animal-abuse | 744 |
| transphobia | 705 |
| abortion | 561 |
| religious-discrimination | 398 |
| ableism | 390 |
| classism | 326 |
| body-shaming | 67 |

Table 3: The 137 Does the Dog Die? trigger warnings used in the ML-DDD dataset along with the number of works marked "Clear Yes" for the warning and number of works marked "Clear No".

| Warning | Works W/ |
|---|---|
| Is there blood/gore | 6121 |
| Is there sexual content | 5406 |
| Is there gun violence | 4588 |
| Does a parent die | 3743 |
| Does someone struggle to breathe | 3372 |
| Does someone abuse alcohol | 3089 |
| Is there a hospital scene | 3074 |
| Is someone kidnapped | 3045 |
| Does it have a sad ending | 2980 |
| Does someone use drugs | 2935 |
| Is there a dead animal | 2822 |
| Is someone tortured | 2762 |
| Does someone fart or spit | 2756 |
| Does a car crash | 2747 |
| Is someone stalked | 2734 |
| Is there shaving/cutting | 2630 |
| Does someone cheat | 2617 |
| Is someone sexually assaulted | 2557 |
| Does an animal die | 2495 |
| Are there flashing lights or images | 2395 |
| Does someone vomit | 2373 |
| Is there hate speech | 2370 |
| Are there anxiety attacks | 2293 |
| Is there shakey cam | 2215 |
| Is a child abused | 2169 |
| Does a kid die | 2167 |
| Are there jumpscares | 2161 |
| Is there domestic violence | 2145 |
| Does someone break a bone | 2076 |
| Is someone restrained | 2027 |
| Is someone gaslighted | 2009 |
| Are needles/syringes used | 1961 |
| Is there a claustrophobic scene | 1925 |
| Is there audio gore | 1912 |
| Is there a shower scene | 1888 |
| Is there addiction | 1864 |
| Does someone die by suicide | 1840 |

| Warning | Works W/ | Warning | Works W/ |
|---|---|---|---|
| Does a car honk or tires screech | 1734 | Is there body dysmorphia | 516 |
| Does someone fall to their death | 1701 | Does someone attempt suicide | 513 |
| Are there bugs | 1700 | Does someone have a seizure | 494 |
| Are animals abused | 1599 | Is there antisemitism | 485 |
| Is there ableist language or behavior | 1599 | Does someone wet/soil themselves | 467 |
| Is someone burned alive | 1508 | Is there a large age gap | 465 |
| Is someone hit by a car | 1506 | Is a child's toy destroyed | 437 |
| Is someone sexually objectified | 1342 | Is there cannibalism | 436 |
| Does the dog die | 1298 | Does a pet die | 431 |
| Is there amputation | 1292 | Are there "Man in a dress" jokes | 420 |
| Is there finger/toe mutilation | 1243 | Does someone fall down stairs | 410 |
| Are there homophobic slurs | 1238 | Does a cat die | 392 |
| Does someone self harm | 1229 | Is the fourth wall broken | 385 |
| Are there fat jokes | 1208 | Does someone sacrifice themselves | 379 |
| Are there nude scenes | 1114 | Are there clowns | 379 |
| Does a head get squashed | 1109 | Is someone buried alive | 376 |
| Is a mentally ill person violent | 1107 | Is someone misgendered | 364 |
| Is there eye mutilation | 1085 | Is the R word used | 360 |
| Does someone drown | 966 | Does someone asphyxiate | 348 |
| Does a baby cry | 965 | Does the black guy die first | 348 |
| Are there babies or unborn children | 944 | Is there copaganda | 342 |
| Is there excessive gore | 907 | Is a minor sexualized | 332 |
| Is there a hanging | 884 | Does a pregnant person die | 293 |
| Are there ghosts | 867 | Is there body horror | 281 |
| Is there misophonia | 834 | Does someone have a chronic illness | 273 |
| Is there obscene language/gestures | 791 | Is there a nuclear explosion | 266 |
| Does someone suffer from PTSD | 785 | Is someone crushed to death | 263 |
| Are any teeth damaged | 782 | Does someone have an eating disorder | 249 |
| Is someone possessed | 721 | Does someone miscarry | 243 |
| Is there a mental institution scene | 712 | Is a minority is misrepresented | 212 |
| Are there snakes | 668 | Is electro-therapy used | 209 |
| Are there n-words | 630 | Is a male character ridiculed for crying | 207 |
| Does a plane crash | 620 | Is a child abandoned by a parent | 191 |
| Are there spiders | 602 | Is someone raped onscreen | 190 |
| Is someone homeless | 598 | Are there abortions | 182 |
| Does someone becomes unconscious | 576 | Does someone overdose | 166 |
| Is there childbirth | 574 | Are there razors | 147 |
| Does an LGBT person die | 544 | Is there autism specific abuse | 124 |
| Is there genital trauma/mutilation | 539 | Is an infant abducted | 112 |
| Does someone have cancer | 539 | Is someone drugged | 105 |
| Are there incestuous relationships | 539 | Is there dog fighting | 105 |
| Does someone say "I'll kill myself" | 524 | Is Santa (et al) spoiled | 87 |

**Continued from previous column**

| Warning | Works W/ |
|---|---|
| Are there mannequins | 83 |
| Does a dragon die | 74 |
| Does someone have a stroke | 68 |
| Is there dementia/Alzheimer's | 53 |
| Is there Achilles Tendon injury | 52 |
| Is a baby stillborn | 50 |
| Are there 9/11 depictions | 38 |
| Are there fat suits | 20 |
| Is there bisexual cheating | 17 |
| Is there D.I.D. misrepresentation | 14 |
| Does a non-human character die | 13 |
| Does the abused become the abuser | 6 |
| Is there bestiality | 4 |
| Is there body dysphoria | 4 |
| Is there aphobia | 3 |
| | Concluded |

# C  Further Details on Recommendation Algorithms

For ML-DDD, we utilize the entirety of the dataset (22.8M interactions) for conducting accuracy evaluations and generating recommendations for 1,000 random users to analyze amplification. For the evaluation of the AO3 dataset (i.e. calculating accuracy metrics), we use a representative subset of the data with 10% of the works and their interactions (3.9M interactions) due to computational constraints. However, we do use the complete dataset to train a model to generate recommendations for 1,000 random users and perform the amplification analysis.

For SVD, the algorithm's factors, epochs, learning rate, and regularization were tuned to optimize rating prediction Root Mean Squared Error (RMSE) on a 90-10 train-test split. For Implicit matrix factorization, the algorithm's factors, regularization, and weight of positive samples are tuned to optimize recall at k=50 recommendations. Hyper-parameters for both datasets can be found in the included code.

## C.1  Evaluation Metrics

To evaluate the performance of the algorithms, we draw from the standard predictive and classification accuracy metrics (Herlocker et al., 2004). For predictive accuracy, we report the Root Mean Squared Error (RMSE) of SVD, the only algorithm which produces rating predictions. RMSE measures the square root of the average squared difference between the 0.5-5.0 scale ratings predicted by the algorithm and the true ratings for user-movie interactions. We split the rating data into a 90-10 train-test split and report the RMSE of the predictions on the test set. The split is made with respect to each user: 90% of the user's interactions are in the trainset, and 10% in the testset.

However, the interaction data in AO3 is implicit: a 1 indicates a user liked an item, and a 0 indicates a user either does not like or has not seen an item. Rather than predicting this binary interaction, the algorithms trained on the AO3 data give recommendations directly. The TopPop and Random algorithms for ML-DDD give recommendations directly as well. To evaluate these algorithms, we report the classification accuracy metrics Precision, Recall, and F1 Metric at k recommendations for three values of k. We adhere by the definitions of these metrics in (Herlocker et al., 2004), summarized here:

- $Precision@k$. Fraction of relevant items in the k recommendations divided by k.
- $Recall@k$. Fraction of relevant items in the k recommendations divided by the total number of relevant items.
- $F1@k$. Two times the product of $Precision@k$ and $Recall@k$ divided by the sum of $Precision@k$ and $Recall@k$.

To define relevancy and calculate the classification metrics, we use a methodology similar to Sarwar et al. (2000); Wilson et al. (2014). We split the rating data into a 90-10 train-test split with respect to each user: again, 90% of the user's interactions are in the trainset, and 10% in the testset. After training each algorithm on the trainset, we generate k recommendations for each user that are unseen in their trainset interactions. The recommendations are then evaluated for precision and recall against the user's relevant testset interactions. A relevant work in AO3 is one that appears in a user's history, as all their interactions are positive. For ML/DDD, we consider movies in user's history that are rated in their top-quartile of interactions as relevant, following suit with Basu et al. (1998).

## C.2  Results

*Predictive Accuracy.* The SVD algorithm has a RMSE of 0.7625 when trained then tested on the 90-10 random split of the data as described in Section 4.2. To generate novel recommendations for the 1,000 randomly selected users in the ML-DDD dataset, we train SVD on the entire dataset and evaluate its performance on the interactions of the random users. This produced a RMSE of 0.6733.

*Classification Accuracy.* The Precision, Recall, and F1 metrics at k = 10, 50, and 100 recommendations are displayed in Table 4. The Random algorithms perform significantly worse than both personalized recommenders on all metrics, which is expected of a simple baseline which does not consider any aspects of the data. In ML-DDD, the Random algorithm slightly outperforms TopPop, which may be due to the high interaction density of the dataset. Conversely, the TopPop algorithm

Table 4: Classification Accuracy Results for each algorithm.

| | Precision @k=10 | Precision @k=50 | Precision @k=100 | Recall @k=10 | Recall @k=50 | Recall @k=100 | F1 @k=10 | F1 @k=50 | F1 @k=100 |
|---|---|---|---|---|---|---|---|---|---|
| **ML-DDD Random** | 1.00e-4 | 1.20e-4 | 4.00e-5 | 5.00e-4 | 2.67e-3 | 2.17e-3 | 1.67e-4 | 2.30e-4 | 7.85e-5 |
| **ML-DDD TopPop** | 1.00e-4 | 2.00e-5 | 2.00e-5 | 5.00e-4 | 5.00e-4 | 7.50e-4 | 1.67e-4 | 9.10e-5 | 9.40e-5 |
| **ML-DDD SVD** | 2.10e-3 | 4.40e-4 | 2.20e-4 | 8.48e-3 | 8.82e-3 | 8.82e-3 | 3.37e-3 | 8.38e-4 | 4.29e-4 |
| **AO3 Random** | 4.64e-5 | 4.96e-5 | 4.86e-5 | 3.17e-4 | 1.71e-3 | 3.29e-3 | 8.09e-5 | 9.64e-5 | 9.59e-5 |
| **AO3 TopPop** | 4.82e-4 | 7.26e-4 | 6.69e-4 | 1.43e-3 | 5.99e-3 | 1.07e-2 | 7.22e-4 | 1.29e-3 | 1.26e-3 |
| **AO3 ALS** | 4.27e-4 | 1.00e-4 | 5.14e-5 | 2.13e-3 | 2.51e-3 | 2.57e-3 | 7.11e-4 | 1.93e-4 | 1.01e-4 |

performs quite well in AO3, outdoing both the Random and ALS algorithms. This indicates the users of AO3 are more likely to have interacted with popular content. SVD achieves higher accuracy than ALS, potentially due to the more telling explicit interactions of ML-DDD.

