# OpenReview forum: "Datasets for Navigating Sensitive Topics in Peference Data and Recommendations"
_NeurIPS.cc/2024/Workshop/SafeGenAi — SafeGenAi Poster_

### Official Review · Reviewer_bTSP · 2024-10-08
**Peer Review for Datasets for Navigating Sensitive Topics in Peference Data and Recommendations**

**Rating:** 5
**Confidence:** 4

**Review:**

## Strengths
1. **Originality**: the originality of the paper is mainly the introduction of the two datasets, ML-DDD and AO3, for studying the sensitive content in recommendation systems.
2. **Study**: the paper also examines the extent of which sensitive content warnings are amplified by different recommendation algorithms, such as TopPop and SVD, compared to random assignments.

## Areas for improvement
1. How should the study results that personalized algorithms do not amplify sensitive content be interpreted?
2. There is opportunity to include more diverse and recent recommendation algorithms, such as neural network-based and graph-based models, to improve the quality of the study. Do they anticipate different amplification behaviors from such models?
3. The paper could benefit from a fuller discussion on the potential bias in sensitive content labels. For example, there are cultural and regional differences in what's considered "sensitive" and how would that impact datasets' generalizability?

---

### Official Review · Reviewer_88TJ · 2024-10-09
**Navigating Sensitive Content: Novel Datasets Expose Recommendation Systems' Impact on Content Sensitivity Exposure.**

**Rating:** 8
**Confidence:** 4

**Review:**

This paper "Datasets for Navigating Sensitive Topics in Preference Data and Recommendations" introduces novel datasets integrated with sensitivity labels to evaluate the impact of recommendation systems on user exposure to sensitive or harmful content. This study addresses an important gap by augmenting MovieLens data with content warnings from "Does the Dog Die?" and fan-fiction interactions from "Archive of Our Own" with user-generated warnings, providing a foundation for understanding how recommendation systems might amplify or mitigate exposure to sensitive content. Through comprehensive statistical analyses and experimental evaluations on these datasets, the authors demonstrate how different recommendation algorithms influence the visibility of content with sensitivity labels. The findings suggest that personalized algorithms do not necessarily amplify sensitive content, posing significant implications for developing more nuanced AI systems that respect user content preferences and sensitivities. This paper is a crucial step towards more ethically aware recommendation systems, offering substantial resources for future research in this area.